# Expediting hit-to-lead progression in drug discovery through reaction prediction and multi-dimensional optimization

David F. Nippa [1,4], Kenneth Atz [1,4], Yannick Stenzhorn [1], Alex T. Müller [1], Andreas Tosstorff [1], Jörg Benz[1], Hayley Binch[1], Markus Bürkler[1], Achi Haider[1], Dominik Heer[1], Remo Hochstrasser[1], Christian Kramer[1], Michael Reutlinger [1], Petra Schneider[2], Thierry Shema[3], Andreas Topp[1], Alexander Walter[1], Matthias B. Wittwer [1], Jens Wolfard [1], Bernd Kuhn [1], Mario van der Stelt [3], Rainer E. Martin [1] ✉, Uwe Grether [1] ✉ & Gisbert Schneider[2] ✉

The rapid and economical synthesis of novel bioactive compounds remains a hurdle in drug discovery efforts. This study demonstrates an integrated medicinal chemistry workflow that effectively diversifies hit and lead structures, enabling an acceleration of the critical hit-to-lead optimization phase. Employing high-throughput experimentation (HTE), we generated a comprehensive data set encompassing 13,490 novel Minisci-type C-H alkylation reactions. These data served as the foundation for training deep graph neural networks to accurately predict reaction outcomes. Scaffold-based enumeration of potential Minisci reaction products, starting from moderate inhibitors of monoacylglycerol lipase (MAGL), yielded a virtual library containing 26,375 molecules. This virtual chemical library was evaluated using reaction prediction, physicochemical property assessment, and structure-based scoring, identifying 212 MAGL inhibitor candidates. Of these, 14 compounds were synthesized and exhibited subnanomolar activity, representing a potency improvement of up to 4500 times over the original hit compound. These ligands also showed favorable pharmacological profiles. Co-crystallization of three computationally designed ligands with the MAGL protein provided structural insights into their binding modes. This study demonstrates the potential of combining miniaturized HTE with deep learning and optimization of molecular properties to reduce cycle times in hit-to-lead progression.

The fast and efficient synthesis of novel compounds continues to be a bottleneck in small-molecule drug discovery, requiring considerable time and effort[1]. The structural novelty and complexity of target molecules for synthesis often present substantial challenges, especially when establishing structure-activity relationships in medicinal chemistry[2]. Therefore, efficient synthetic strategies are crucial for hit-to-lead and lead optimization programs, as they play a key role in improving or maintaining pharmacological activities while

[1]Roche Pharma Research and Early Development (pRED), Roche Innovation Center Basel, F. Hoffmann-La Roche Ltd., Grenzacherstrasse 124, Basel, Switzerland. [2]ETH Zurich, Department of Biosystems Science and Engineering, CURE, Klingelbergstrasse 48, Basel, Switzerland. [3]Leiden University, Leiden Institute of Chemistry, Department of Molecular Physiology, Einsteinweg 55, CC, Leiden, The Netherlands. [4]These authors contributed equally: David F. Nippa, Kenneth Atz. ✉e-mail: rainer_e.martin@roche.com; uwe.grether@roche.com; gisbert@ethz.ch

concomitantly improving the physicochemical properties of drug candidates[3]. Advanced computer-assisted molecular design methods have the potential to address some of these key aspects in medicinal chemistry[4].

Using computational tools and machine learning frameworks, combined with efficient semi-automated synthesis techniques such as high-throughput experimentation (HTE) and late-stage functionalization (LSF), has been shown to improve cycle times while reducing material consumption and costs[5,6]. LSF facilitates the direct substitution of C-H bonds by other functional groups or building blocks in a single step, eliminating the need for pre-functionalization of complex molecules[7]. Consequently, this approach can offer higher efficiency in diversifying hit and lead structures compared to traditional multi-step synthesis starting from basic building blocks[8]. Among the available LSF methodologies[9], Minisci-type C-H alkylations are of particular interest as they leverage readily available carboxylic acids to incorporate alkyl fragments into heterocyclic cores[10], such as the widely used pyridine motif in medicinal chemistry. In addition, the use of carboxylic acid building blocks with many carbon $sp^3$ centers can improve lead- and drug-like characteristics, activity and selectivity profiles, as well as the physicochemical molecular properties of candidate compounds[11,12].

However, due to the structural intricacies and the number of similar C-H bonds in a molecule, the transformation of pharmaceuticals and advanced building blocks using LSF reactions remains challenging[8,9]. HTE partially alleviates these issues by enabling the rapid identification and optimization of reaction conditions with minimal material consumption[6,13,14]. The curation of the generated HTE data based on the FAIR principles (findable, accessible, interoperable, and reusable)[15,16] can help create high-quality data sets suitable for computational analyses and machine learning applications[17,18]. We previously reported successful in silico prediction of the coupling performance of the coupling of heterocyclic building blocks with diverse $sp^3$-rich carboxylic acids using Minisci-type reactions, using graph neural networks (GNN) trained on HTE data[19]. GNNs enable learning on two-dimensional (2D) and three-dimensional (3D) molecular models[20–22]. These deep learning models were developed to predict reaction outcomes, including product yields and regioselectivity, such as for borylation and alkylation reactions[5,23–25]. Recent studies have shown how GNNs can be used for the optimization of conditions in cross-coupling reactions[26], and reaction outcome prediction for three-component reactions[27].

Hit-to-lead optimization in drug discovery projects typically combines optimization of on-target activity and drug selectivity against undesired off-targets with achieving suitable absorption, distribution, metabolism, and excretion (ADME) properties[28–30]. Although machine learning models have shown robust prediction accuracy for certain physicochemical properties, such as lipophilicity or permeability[31,32], predicting drug potency or protein-ligand complexes proves to be notoriously challenging[33,34]. The methods used include physics-based approaches, such as free energy perturbation calculations[35], semi-empirical quantum chemistry approximations[36,37], and deep learning, including GNN approaches[38,39]. Moreover, ligand-based machine learning was combined with three-dimensional (3D) scoring using template-based ligand docking[40].

This study describes an efficient compound diversification strategy aimed at enhancing the potency of existing MAGL-inhibiting molecules (hits). We combine GNN-based C-H activation reaction prediction for computational molecular library design with a suite of property prediction methods. These include structure-based scoring using template docking and prediction of key molecular properties such as lipophilicity, solubility, and permeability. Toward this end, a new Minisci alkylation data set with reaction yield resolution was created that covers 13,490 reactions. Integrating a reaction prediction model trained on these data with different scoring methods ("machine learning funnel") enabled the computational generation of novel,

synthetically accessible compound designs with improved physico-chemical properties and potency. We applied this processing cascade to monoacylglycerol lipase (MAGL), a serine hydrolase that is crucial for the metabolism of neuroprotective endocannabinoid 2-arachidonoylglycerol (2-AG). MAGL represents a target for the development of selective inhibitors aimed at treating central nervous system (CNS) disorders associated with neuroinflammation[41–43]. Using a selected set of moderate MAGL-inhibitors (hit compounds) that allow diversification through C-H Minisci-type alkylation in combination with a set of 211 carboxylic acids, we generated an enumerated data set of 26,375 in silico reaction products (Fig. 1). These virtual molecules were then subjected to the machine learning funnel, delivering 212 diversified, novel, synthetically accessible designs with predicted improved physicochemical properties and potency, of which 14 were synthesized and tested in biochemical assays for inhibition of human, mouse, rat, and macaque MAGL enzymes.

## Results

### Reaction data set generation with miniaturized high-throughput experimentation

Generation of the reaction data set for Minisci-type alkylation reactions involved the experimental evaluation of a variety of electron-deficient heterocyclic fragments in combination with various $sp^3$-rich carboxylic acids. 80 common medicinal chemistry fragments (*e.g.*, pyridines, pyrimidines, quinolines, benzoxazoles, **F1-F80**, Supplementary Information [SI]4.1) and 59 carboxylic acids (**A1-A59**, SI4.2) were selected, resulting in a chemical space of 4,720 possible combinations (Fig. 2A). Data analysis using the Simple User-Friendly Reaction Format (SURF)[16] revealed variable reactivity dependent on different silver salts and additives while maintaining consistent oxidant and solvent systems. Consequently, each fragment+acid combination was assessed with different catalysts and additives in a 24-well plate format (Fig. 2F, SI3).

The theoretical reaction space generated by the substrates and conditions in focus encompassed nearly 115,000 possible combinations. To allow efficient evaluation of a relevant portion of this reaction space, a rapid and miniaturized HTE workflow was implemented (Fig. 2B, SI3). Using stock solutions and automated liquid handling, we prepared reaction plates in a reproducible and efficient manner. Reactions were analyzed via liquid chromatography-mass spectrometry (LC-MS) to determine the reaction outcome, employing an automated tagging workflow. In total, 13,490 reactions were performed, representing approximately 12% of all possible combinations (Fig. 2C). Data analysis revealed that 30% of the reactions resulted in positive reaction outcomes, *i.e.*, successful alkylation of the heterocyclic fragment with a detectable reaction yield of ≥5% (Fig. 2D). Among these successful reactions, a balanced yield distribution was observed (Fig. 2E), making this generated data set suitable for training a machine learning model for reaction forward prediction.

### Building reaction forward prediction models based on the Minisci data set

Machine learning models were trained to predict reaction outcomes using various data set splitting strategies (Fig. 3A), including a 0D split for known combinations, 1D splits for novel acids or *N*-arenes, and a 2D split for novel combinations of both components (Fig. 3B). The performance of these models, assessed through reaction yield and binary outcome predictions, highlights their capacity for reaction forward prediction (Methods and SI1). The random split (0D) consistently outperformed the other strategies, with a mean absolute error (MAE) of 6.7% and a Pearson correlation coefficient (*r*) of 0.83, demonstrating predictive power for both reaction yield and binary outcomes (accuracy = 85.7%, precision = 90.9%) (SI1). Extrapolation to novel acids (1DA) and novel *N*-arenes (1DN) resulted in increased errors, with MAEs of 12.5% and 11.4%, respectively. The 2D split, which tested novel

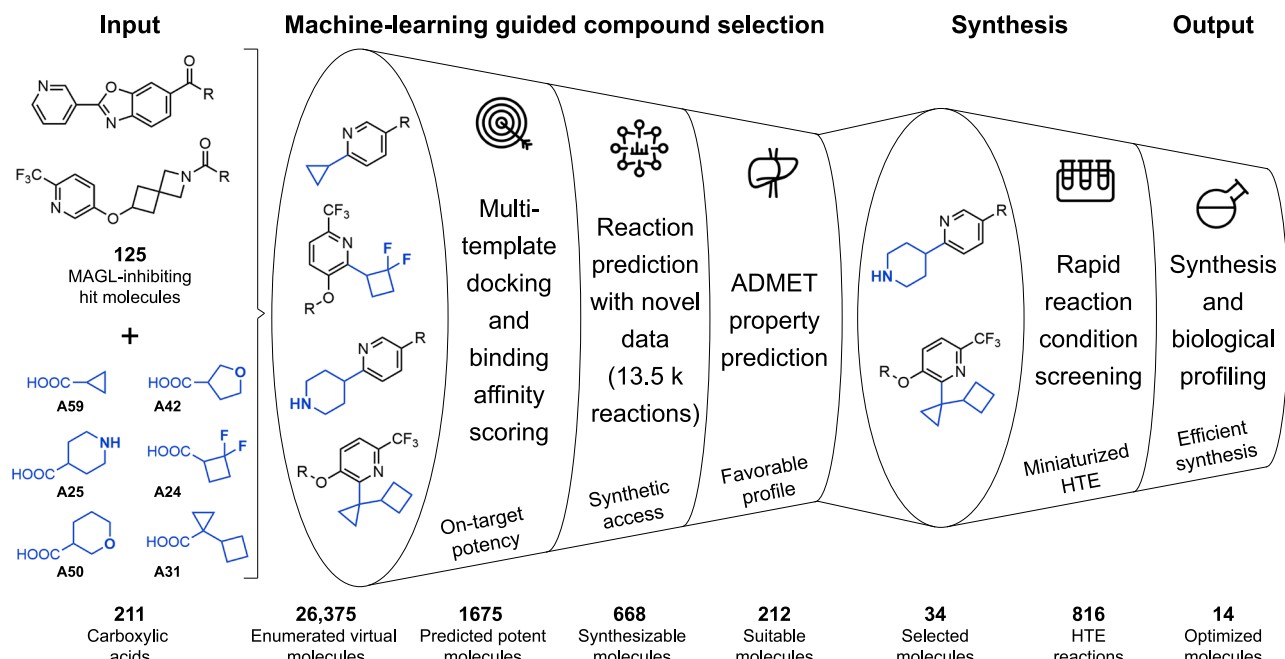

**Fig. 1 | Hit optimization workflow.** A set of 125 monoacylglycerol lipase (MAGL)-inhibiting "hit" molecules (starting points) containing *N*-heterocycles are combined with 211 commercially available carboxylic acids to generate 26,375 hypothetical products. These virtual molecules are passed through a series of filters to identify potent, synthetically accessible molecules with favorable physicochemical properties ("machine learning funnel"). Multi-template docking and a machine learning-based potency scoring function are used to identify candidate molecules. Synthetic accessibility is predicted using graph neural networks (GNNs) trained on a novel reaction data set encompassing 13,490 Minisci-type alkylation reactions. Physicochemical properties are determined using various, readily available machine learning-based predictions. Consequently, from 212 potentially suitable compounds, 34 molecule-acid combinations were manually selected and screened using miniaturized high-throughput experimentation (HTE) to identify favorable alkylation reaction conditions. Based on building block availability, up-scaling of a selected number of reactions successfully delivered 14 compounds with improved profiles.

combinations of both acids and *N*-arenes, showed comparable performance to the 1D splits, with an MAE of 12. 6%, maintaining a relatively high precision at 85.4% (SI1).

## Molecular template docking, synthesizability assessment, and property prediction for candidate selection

To explore the potential chemical space using C-H alkylation, 125 hit molecules featuring *N*-heteroaromatic scaffolds and moderate MAGL inhibitory activity were selected. These molecules were combined with 211 carboxylic acid fragments characterized by a high sp$^3$ content, covering a virtual library of 26,375 potential products. The compounds were subsequently evaluated using a multi-dimensional optimization (MDO) strategy that included protein-ligand template docking, synthetic feasibility assessment using reaction prediction, as well as physicochemical and ADME property prediction (lipophilicity calculated as LogD values, kinetic solubility assay (LYSA), P-glycoprotein (P-gp) apparent permeability, and parallel artificial membrane permeability assay (PAMPA)). For MAGL, an intracellular target, the cellular permeability of inhibitors is a crucial factor for successful drug development. This MDO approach allowed for the simultaneous evaluation of diverse criteria, ensuring the identification of compounds that balance potency, synthesizability, and properties that impact their pharmacokinetic profiles. Figure 1 illustrates how the applied computational filtering process enabled prioritization and selection of compounds with predicted increased inhibitor potency and synthetic accessibility compared to the original hits.

A subset of 1675 molecules with predicted inhibitory potencies of pIC$_{50}$ ≥6 was identified, using a recently published 2D3D-hybrid scoring method[40] (Fig. 3C). The application of the GNN reaction prediction model to validate the synthetic feasibility resulted in 668 compounds with a predicted reaction yield ≥5% (Fig. 3D), confirming a positive, binary reaction outcome. From this pool, 212 molecules had both a predicted positive reaction outcome (*i.e.*, ≥5% LCMS yield) and a pIC$_{50}$ ≥8, which made them preferred candidates for further investigation (Fig. 3E). The computational docking approach used fixed coordinates of a predefined template as illustrated in Fig. 3F overlayed with 10 potential candidate ligands. Finally, the predictions of four selected physicochemical properties (*i.e.*, LogD, LYSA, P-gp, PAMPA) of the remaining 212 molecules were evaluated (Fig. 3G), leading to the selection of 34 molecule+carboxylic acid combinations exhibiting a balance in key physicochemical properties and ADME parameters. The reaction and potency predictions served as strict criteria, narrowing the virtual library to 212 molecules. The final selection of 14 compounds was guided by calculated ADME parameters, using an approach that avoided fixed thresholds to minimize the propagation of model error through sequential filtering steps.

## MAGL inhibitor synthesis and biological characterization

The 34 identified molecules were subjected to screening of reaction conditions using a 24-well setup (Fig. 2F) resulting in the formation of the desired product with ≥5% for the 34 selected substrate combinations. Based on these results, the most high-yielding examples were prioritized and, consequently, scale-up experiments were performed for 14 new MAGL inhibitors (**18-31** following the synthesis route outlined in Fig. 4A. Detailed synthesis information can be found in SI5. Using our late stage Minisci-type building block alkylation procedure, we reduced the number of synthesis steps of the alkylated intermediate from the seven steps proposed by a classical synthesis approach (Fig. 4A, orange) to three steps (Fig. 4A, blue).

The inhibitory potency of all newly synthesized compounds (**18-31**) was evaluated in a biochemical assay for the human, mouse, rat and cynomolgus macaque MAGL enzymes. Compared to the human IC$_{50}$ value of the starting point, hit compound **17** (IC$_{50}$ = 445 nM), all newly synthesized compounds exhibited enhanced potency (Table 1). Six

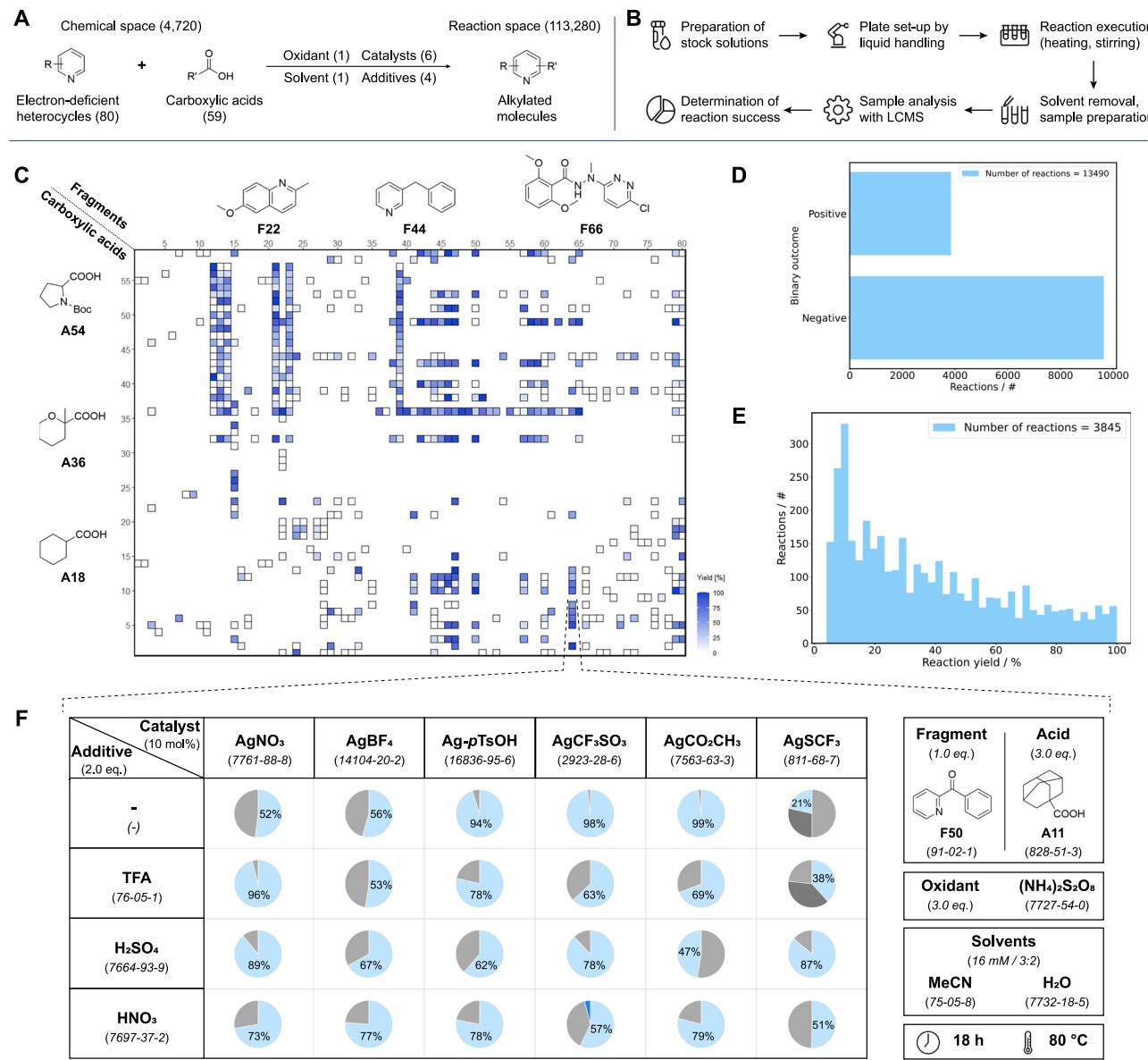

**Fig. 2 | High-throughput generation of experimental training data for reaction forward prediction models. A** Generalized Minisci reaction scheme highlighting the large chemical and reaction space. **B** Miniaturized reaction screening workflow utilized to generate the data set. **C** Matrix of possible fragment and carboxylic acid combinations including conducted screenings and their yield range. **D** Analysis of the binary reaction outcome. **E** Analysis of the yield across all reactions. **F** Example of the used screening plate highlighting the results from one particular fragment and carboxylic acid combination. Colors of the pie charts: Light blue: Mono-alkylation product, Grey: Starting material, Dark blue: Di-alkylation product, Dark grey: Side products. The percentage values represent the amount of observed mono-alkylated product determined by liquid chromatography-mass spectro-metry (LCMS).

compounds (**21**, **23**, **24**, **27**, **29**, and **31**) demonstrated $IC_{50}$ values between 0.1 and 10 nM, *i.e.*, corresponding to a 45- to 4500-fold potency enhancement, with four molecules (**25**, **26**, **29** and **31**) achieving considerably improved lipophilic efficiency (LipE), *i.e.*, = $pIC_{50}$ - cLog*P*. The $IC_{50}$ values predicted by the 2D3D-hybrid scoring model reflect the trend of the experimentally confirmed potency with a Pearson correlation of $r = 0.52$.

To confirm the ligand binding mode, the crystal structures of four inhibitor-MAGL complexes (**17**, **23**, **27**, and **29**) were determined (Fig. 4B). These co-crystal structures reveal that the addition of cyclopentyl and cyclopropyl-cyclobutyl groups in the *para* position of the pyridine induces a flip of the pyridine, allowing a new hydrogen bond interaction with a water molecule and allowing lipophilic resi-dues to access a previously untapped sub-pocket of the MAGL binding site. The interactions on the right-hand side of the ligands, including key hydrogen bonds to Met[123], Ala[51], Arg[57], two crystal water molecules,

and π-stacking with Tyr[194], remain conserved compared to the original hit structure **17**.

For the three compounds (**23**, **27**, and **29**), additional physico-chemical and ADMET properties were obtained (SI2.2). The high selectivity toward MAGL over common safety-relevant off-targets and closely related brain hydrolases (SI2.3) was confirmed by activity-based protein profiling[43] (SI2.4).

## Discussion

Curated high-quality reaction data sets are the cornerstone of a suc-cessful forward reaction prediction model. The data used for machine learning in this study consisted of one Minisci-type alkylation data set with 13,490 transformations covering a wide range of scaffolds rele-vant to medicinal chemistry. Data points were generated in a resource-efficient and time-efficient HTE setup, which required minimal starting material. This experimental approach allowed us to explore a relevant

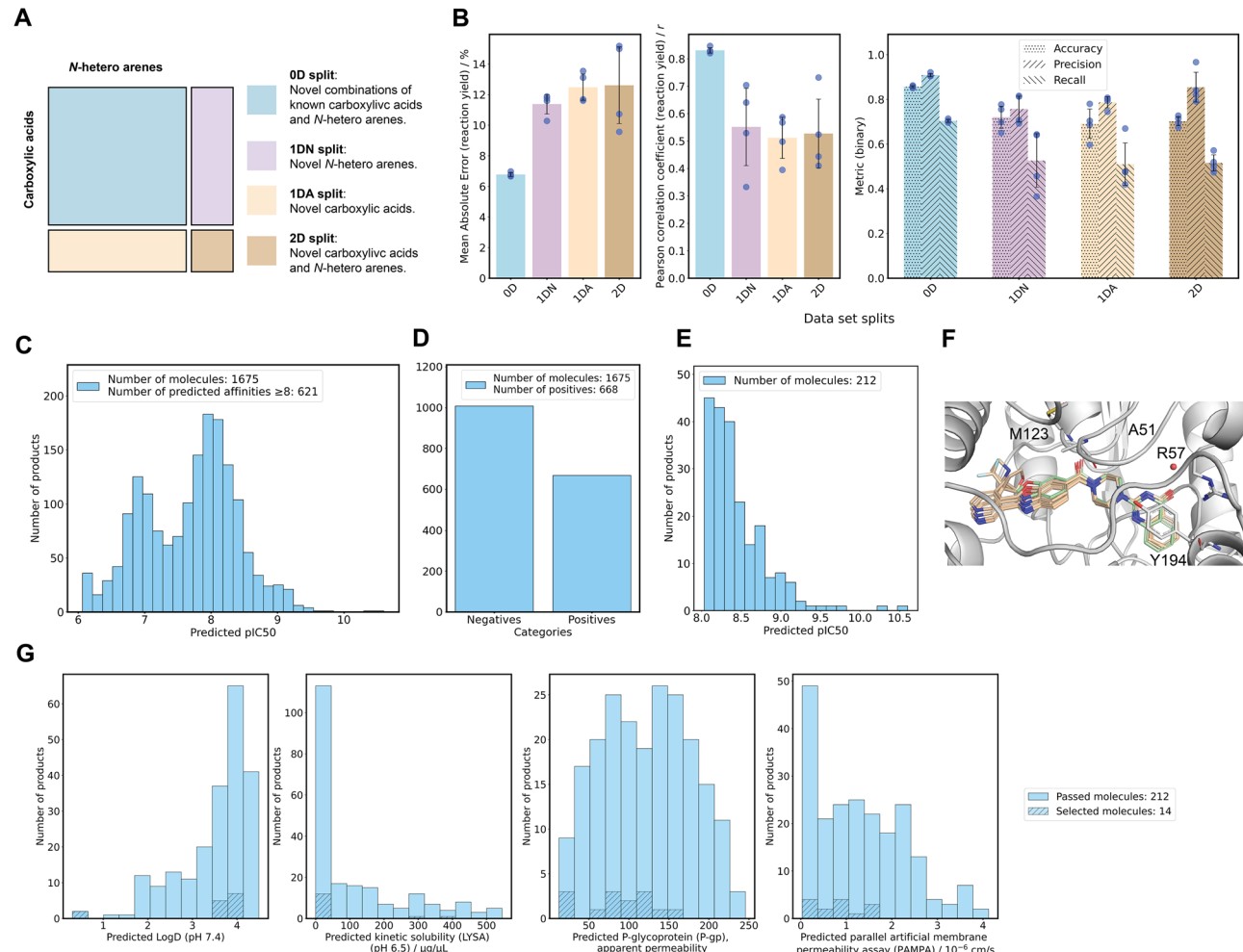

**Fig. 3 | Computational filtering and selection of molecules. A** To test the extrapolation performance of the machine learning models, the reaction data set was split using four different strategies: (i) Predicting novel combinations between known *N*-arenes and acids (zero-dimensional [0D] split, shown in light blue); (ii) Extrapolation to novel *N*-arenes (one-dimensional [1D] split for *N*-arenes [1DN], shown in pale purple); (iii) Extrapolation to novel acids (1D split for acids [1DA], shown in beige); and (iv) Extrapolation to both novel *N*-arenes and acids (2D split, shown in light brown). **B** Visualization of machine learning results for the four data set splitting strategies, showing reaction yield prediction (left and center) and binary reaction outcomes (right). Error bars indicate the standard deviation of a four-fold cross-validation; individual data points are shown. **C** Predicted potency for generated molecules with a predicted pIC$_{50}$≥6. **D** Predicted reaction outcomes: Molecules with a predicted reaction yield ≥5% are classified as positive, otherwise

negative. **E** Visualization of the subset of molecules with predicted positive reaction outcomes and pIC$_{50}$ ≥8. **F** Template docking was demonstrated with an example template and four products, where the coordinates of the template were kept fixed. Key amino acids are shown, *i.e.*, Met[123], Ala[51], Arg[57], and Tyr[194]. The template compound is illustrated in light green indicating the initial position and placement of the docked ligands and the four products in the docked conformation in light brown. **G** Predicted absorption, distribution, metabolism and excretion (ADME) properties of the final subset of 212 molecules. The physicochemical and ADME properties considered are (from left to right): LogD, kinetic solubility assay (LYSA) solubility in μg/mL, P-glycoprotein (P-gp) apparent permeability, and parallel artificial membrane permeability assay (PAMPA) in 10$^{-6}$cm/s. The 14 molecules selected for synthesis are indicated by dashed lines. pIC$_{50}$ = The negative logarithmic concentration of the half maximal inhibitory concentration in *mol/L*.

portion (12%) of the reaction space, encompassing approximately 115,000 potential combinations. This sampling was sufficient for robust machine learning model training. To make the data available for reaction prediction, they were exported in SURF[16]. SURF files provide reaction information in a comprehensive machine-readable format containing all parameters of the transformation, as well as structural and quantitative information of all components and products. Similarly to related recent work[27,44], it became evident that such machine-readable data sets, including negative reaction outcomes, are the key enablers for successful reaction prediction.

The reaction prediction models demonstrated their robustness not only by accurately predicting the reaction outcome for known substrates, as seen with the random data split (0D split), which achieved a mean absolute error (MAE) of approximately 7%, high correlation (*r*=0.83), and a precision value of 91%. When applied to unseen carboxylic acids and N-arenes, either independently (1D split)

or combined (2D split), the models maintained robust performance, with MAEs in the range of 11-13%, precision values up to 85%, and correlation coefficients ranging from *r* = 0.51 to *r* = 0.55. This ability to partially extrapolate to novel chemical space, combined with the integration of predicted synthetic feasibility, provides an advantage over relying solely on the enumeration and screening of large virtual libraries, where synthetic accessibility is often a downstream challenge[27].

We applied multi-template ligand docking to a virtual library of 26,375 potential compounds originating from MAGL-inhibiting hit molecules, narrowing the pool to 1675 candidates based on predicted potency with pIC$_{50}$ ≥6. The reaction prediction models were then used to assess the synthetic feasibility of these candidates (binary "yes/no" reaction outcome), identifying 668 molecules with predicted reaction yields of ≥5%. In the final step, a physicochemical and ADMET property prediction funnel was used to automatically select 34 molecules with

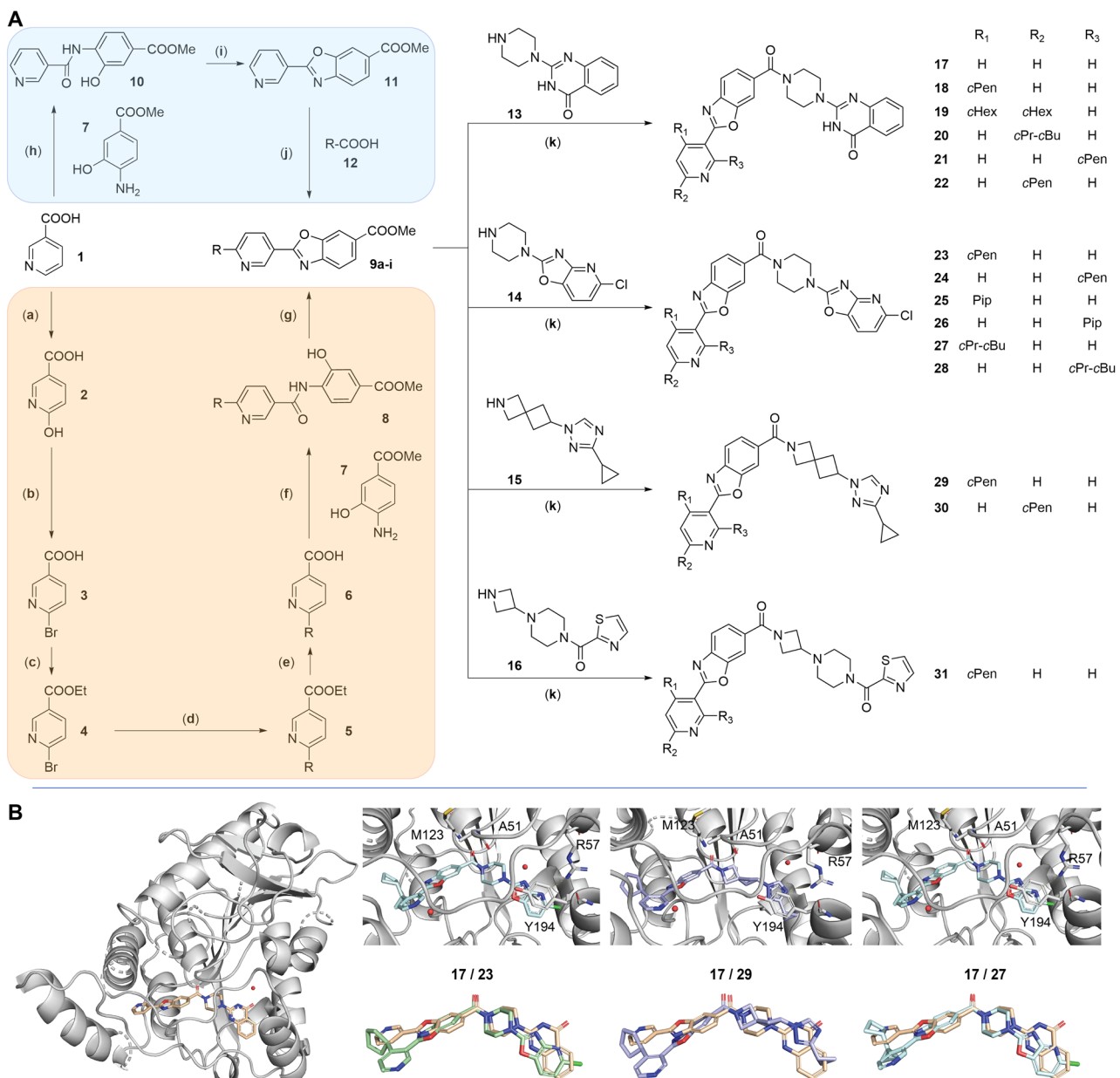

**Fig. 4 | Synthesis route to novel MAGL inhibitors. A** The Minisci-type alkylation approach delivered a shortened synthesis route. Rather than through seven steps (orange, **a–g**), the left-hand side building blocks (**9a-i**) were obtained in three steps (blue, **h–j**). Consequent coupling with different right-hand side building blocks (**13–16**) delivered 14 MAGL inhibitors (**18–31**). **a** $H_2O$, 0 °C, 2 eq. KOH, 10% $F_2$ in $N_2$; **b** neat, 120 °C, 1.4 eq. $PBr_5$; **c** EtOH, rt, 1.0 eq. $NaBH_4$; **d** Toluene/$H_2O$ 10:1, 100 °C, 1.26 eq. R-boronic acid, 0.3 eq. $PCy_3$, 0.1 eq. $Pd_2(dba)_3*CHCl_3$, 3.0 eq. $K_2CO_3$; **e** HCl aq., 80 °C; **f** THF, rt, 1.3 eq. CDI, 1.3 eq. **7**; **g** polyphosphoric acid, 120 °C; **h** THF, rt, 1.3 eq. CDI, 1.3 eq.; **i** polyphosphoric acid, 120 °C; **j** MeCN/$H_2O$ 3:2, 80 °C, 3.0 eq. **12**, 3.0 eq. $(NH_4)_2S_2O_8$, 0.1 eq. $AgSCF_3$; **k** a. MeOH, rt, 5.0 eq. LiOH b. DMF, rt, 1.2 eq. **13/14/15/16**, 10.0 eq. DIPEA, 1.1 eq. HATU; **B** Crystal structure complexes of human monoacylglycerol lipase (MAGL) with hit compound **17**, and inhibitors **23**, **27**, and **29**, shown in pale, light green, light blue, and royal blue, respectively. From left to right, a close-up view of the binding poses of **23**, **27**, and **29**, along with some of their key interactions (*i.e.*, hydrogen bonds to Met[123], Ala[51], Arg[57], and two crystal water molecules, and π-stacking with Tyr[194]) is shown. Additionally, an overlay of the three inhibitors (**23**, **27**, and **29**) with the hit structure **17** is depicted. Source data are provided in a Source Data file. Abbreviations: *c*Pen: Cyclopentyl (**A37**), *c*Hex: Cyclohexyl (**A36**), *c*Pr-*c*Bu: Cyclopropyl-cyclobutyl (**A31**), Pip: Piperidine (**A25**).

favorable solubility and permeability profiles. Subsequently, these top-ranking candidates were prioritized for synthesis and biological testing. This workflow utilized an MDO strategy that allowed the simultaneous consideration of potency, synthetic feasibility, and physicochemical properties. The MDO framework streamlined the prioritization of compounds that achieved a balance across these diverse criteria. This combined approach accelerated the hit-to-lead optimization process by reducing the need for manual intervention and experimental resources. The integration of reaction yield

prediction into this workflow was crucial, highlighting the impact of predictive modeling on synthetic efficiency and reduction of material waste in medicinal chemistry[45]. The complete optimization cycle was performed in one month. The computational work, including virtual library enumeration, multi-template docking, reaction prediction, and ADMET property prediction, was completed in approximately one day. Compound synthesis, using miniaturized HTE and scale-up, took two weeks. The biological measurements were also completed in two weeks. This condensed timeline demonstrates how integrating

**Table 1 | Potency data and lipophilic efficiency (LipE) (i.e., = pIC$_{50}$ - cLogP) of the synthesized monoacylglycerol lipase (MAGL) inhibitors**

| MAGL inhibitor | IC$_{50}$ human/ nM | IC$_{50}$ mouse/ nM | IC$_{50}$ rat/ nM | IC$_{50}$ cynomolgus macaque/ nM | IC$_{50}$ nanoBRET human/ nM | Predicted potency / pIC$_{50}$ & nM | Lipophilic efficiency (LipE) |
|---|---|---|---|---|---|---|---|
| **17** (hit) | 445 | 557 | 79 | 531 | >100 | - | 4.14 |
| **20** | 50 | >10000 | 26 | 61 | 5.1 | 10.3 / 0.050 | 2.84 |
| **21** | 10 | 27 | 2 | 12 | 3.9 | 8.3 / 5.01 | 3.84 |
| **22** | 66 | 189 | 18 | 68 | 19.2 | 9.8 / 0.158 | 3.02 |
| **23** | 4 (±1.3) | 109 (±1.1) | 6 (±1.2) | 5 (±1.1) | 5.9 | 9.5 / 0.316 | 3.50 |
| **24** | 7 (±1.2) | 71 (±1.2) | 2 (±1.1) | 7 (±1.1) | 2.7 | 9.1 / 0.794 | 3.04 |
| **25** | 434 (±1.1) | >10000 | 206 (±1.0) | 581 (±1.6) | >100 | 8.3 / 5.01 | 6.04 |
| **26** | 348 (±1.3) | 6426 | 155 (±1.3) | 547 (±1.2) | >100 | 8.3 / 5.01 | 5.91 |
| **27** | 2 (±1.2) | 86 (±1.5) | 8 (±1.2) | 2 (±1.5) | 14.5 | 10.6 / 0.025 | 3.72 |
| **28** | 19 (±1.3) | 369 (±2.1) | 13 (±1.2) | 28 (±1.3) | 10.8 | 9.4 / 0.398 | 2.31 |
| **29** | 0.1 (±1.6) | 3 (±1.7) | 0.7 (±1.2) | 0.1 (±2.4) | 0.31 | 9.6 / 0.251 | 6.27 |
| **30** | 60 (±1.3) | 253 (±1.4) | 99 (±1.2) | 84 (±1.2) | 11.8 | 8.1 / 7.9 | 3.28 |
| **31** | 0.6 (±1.3) | 0.8 (±1.3) | 1 (±1.1) | 0.8 (±1.2) | 0.1 | 8.9 / 1.18 | 6.21 |

LipE estimates a compound's drug-likeness by integrating both its potency and lipophilicity into a single value. NanoBRET is a proximity-based assay that detects small molecule-binding to a target in live cells using bioluminescence resonance energy transfer (BRET) as described in ref.62. The numbers in the parentheses are ± standard deviation of N = 3 measurements. IC$_{50}$ values were determined in triplicate (N = 3), whereas NanoBRET assays were performed once (N = 1).

computational and experimental workflows from the outset outpaces traditional drug discovery pipelines that do not include predictive modeling.

The 34 selected combinations of molecules and carboxylic acids were subjected to screening of reaction conditions in a 24-well plate setup (Fig. 2F), resulting in the formation of the desired product with ≥5% LCMS-yield for all substrate combinations. Scale-up experiments were performed for 14 new MAGL inhibitors (**18-31**) following the simplified synthesis route outlined in Fig. 4A. By employing a late-stage Minisci-type alkylation strategy, we reduced the synthesis of the alkylated intermediate from seven steps (orange) to three steps (blue), greatly enhancing efficiency. Instead of alkylating the final compound, we synthesized intermediate building blocks, allowing diverse amide couplings to generate a broader array of analogs.

All synthesized compounds (**18-31**) were evaluated for potency using MAGL biochemical assays. Compared to the IC$_{50}$ of the structurally related original hit (**17**), all newly synthesized compounds showed improved potency (Table 1). Six inhibitors demonstrated a 45-fold to a 4500-fold increase in potency, with two compounds, **29** and **31**, achieving subnanomolar activity. The predicted IC$_{50}$ values reflect the trend of the measured biological data (r = 0.52). A LipE analysis revealed that four molecules (**25**, **26**, **29**, and **31**) achieved LipE values greater than 4.1, i.e., the LipE value of the original hit **17**. This value indicates a favorable balance between potency and lipophilicity, which is important for the development of effective and selective inhibitors. Compounds **29** and **31** exhibit potent cell permeability, a crucial characteristic for the intracellular target MAGL, positioning them in the favorable upper right corner of the optimization chart (SI2.2, Fig. S1).

To confirm and study their binding modes, the co-crystal structures of four inhibitors (**17**, **23**, **27**, and **29**) with human MAGL were determined (Fig. 4B). Compared to the binding mode observed for compound **17**, the benzoxazole motif on the left side of the molecules is inverted in the new compounds. This flip is likely driven by additional hydrophobic interactions formed by the new substituents in this region of the MAGL binding site.

The moderate correlation (r = 0.52) of the potency predictions with the experimental results reflects an inherent trade-off between computational cost and accuracy. Free energy perturbation[35] or semi-empirical quantum chemistry approximations[36,37] could potentially refine ΔG estimates for a small subset of ligands but are impractical for large-scale screening. In contrast, deep-learning models enable rapid

scoring of large molecule libraries, yet remain limited by the quality of their scoring functions. Here, we adopted template-based docking with RF-scoring, which has been shown to efficiently enrich active compounds[40]. Developing next-generation scoring functions that combine the fidelity of physics-based methods with the throughput of machine learning models represents an opportunity for in silico hit and lead profiling.

This study presents a comprehensive machine learning-based workflow that integrates binding potency predictions, reaction forward prediction models, and ADMET property evaluation to rapidly select, prioritize, and synthesize new chemical entities with strongly enhanced inhibitory activity. The workflow accelerates the early hit-to-lead optimization cycle by rapidly diversifying and evaluating new molecules, thereby reducing both cycle times and material consumption. This sequential process is adaptable to other projects and reaction types. It provides a distinct advantage over simply enumerating and screening vast virtual libraries, primarily because exhaustive library enumeration often presents considerable synthetic challenges in laboratory experiments. Importantly, the applicability and success of our workflow hinge on the availability of high-quality, well-curated data sets for machine learning and access to reliable binding models for computational template-based docking and affinity prediction. We advocate for the adoption of FAIR principles (Findable, Accessible, Interoperable, Reusable) in data collection throughout the drug discovery cycle, from synthesis to biological evaluation. Such efforts will enable machine learning cascades to further streamline and accelerate hit-to-lead optimization in medicinal chemistry.

## Methods
### Screening plate design
The collection and meta-analysis of reaction data were based on publications covering Minisci-type alkylation using carboxylic acids as radical precursors, as reviewed by Proctor[46] and Duncton[10]. A 24-well alkylation screening plate was designed using reaction conditions most frequently reported in the literature. The reaction variables included temperature, time, compound concentrations, and scale, optimized for minimal material consumption and ease of integration into the experimental workflow. By miniaturizing the scale and adapting the concentration ranges to c = 0.016 M (n = 0.8 μmol), we achieved solid conversion rates for literature test reactions. This was done at a temperature of 80 °C for 18 h. Since most of the reactions worked with ammonium persulfate (($NH_4$)$_2S_2O_8$) as the oxidant, only a

single oxidant was used on the plate. The observed variety in silver salts (catalyst) led to the inclusion of six different species ($AgNO_3$, $AgBF_4$, Ag-pTsOH, $AgCF_3SO_3$, $AgCO_2CH_3$, and $AgSCF_3$). Based on literature analysis and with four additional parameters available per plate, we opted to vary additives rather than solvents. Consequently, TFA, $H_2SO_4$, $HNO_3$, and a non-additive control were selected to allow for a broad range of pKa values. As a solvent system, a mixture of MeCN and $H_2O$ with a 3:2 ratio was chosen (Fig. 2F).

### Generation of Minisci-type alkylation reaction data set with miniaturized HTE

Using the 24-well plate design (Fig. 2F), we evaluated Minisci-type alkylations between a curated set of 80 commercially available *N*-heteroarenes (comprising pyridines, pyrimidines, and indoles, labeled **F1-F80** in SI 4.1) and 59 sp³-rich carboxylic acids (encompassing linear, branched, cyclic, heterocyclic, and aromatic motifs, labeled **A1-A59** in SI 4.2). Candidates were first filtered to remove incompatible functional groups. They were then required to be in stock (greater than 1g) from at least one supplier, and finally, manually down-selected to maximize scaffold diversity. The reactions were set up using automated liquid handling of prepared stock solutions, and then executed in glass vials on a parallel screening plate under a normal atmosphere, with heating and stirring. Upon completion of the reaction, the solvents were removed by evaporation, followed by automated resuspension of the residues in MeCN/$H_2O$ and dilution to a defined concentration for LC-MS analysis using automated liquid handling. The resulting reaction data were subjected to an automated analysis pipeline to rapidly determine all components within the mixture[5]. The standardized output of the data in SURF allowed for direct visualization of the reaction outcome and generated a data set containing 13,490 reactions for training specifically designed GNN-based machine learning models. The general screening procedure, including detailed information on the hardware and software utilized, is provided in SI3.

### Reaction prediction

**Machine learning process.** PyTorch Geometric (2.0.2)[47] and PyTorch (1.10.1+cu102)[48] functionalities were used for GNN training. Training was performed on a GPU (NVIDIA A100 Tensor Core-GPU) graphical processing unit for four hours, using a batch size of 16 samples. We used the Adam stochastic gradient descent optimizer[49], with a learning rate of $10^{-4}$, loss of mean squared error (MSE) on the training set, a decay factor of 0.5 applied after 100 epochs, and an exponential smoothing factor of 0.9. Early stopping was applied to the model that achieved the lowest validation mean absolute error (MAE) in 1000 epochs. All models considered in this study were trained on the Roche high-performance computing cluster, Switzerland.

**Atom featurization.** Atomic properties were encoded *via* the following atomic one-hot encoding scheme: 12 atom types [H, C, N, O, F, P, S, Cl, Br, I, Si, Se], two ring types [True, False], two aromaticity types [True, False], four hybridization types [SP³, SP², SP, S].

**Reaction condition featurization.** Molecular reaction conditions, *i.e.*, additives, solvents, atmosphere, catalysts and reagents, were one-hot encoded. The data set covered four reagents and four solvents, 11 catalysts, four additives, and two atmospheres.

**Conformer generation.** 3D conformers were calculated using RDKit (`AllChem.EmbedMolecule`[50]) followed by energy minimization using the Universal Force Field (UFF) method[51]. For each molecule, ten different conformers were calculated for training and testing. A conformer was randomly selected at each training step. For testing, the final predictions were obtained by averaging the individual predictions calculated for each of the ten conformers.

**Graph neural network architecture.** A graph transformer neural network (GTNN) architecture was employed based on the E(3) equivariant graph neural network (EGNN) architecture[52], which has been used in several related applications[38,53]. The GTNN was designed using the same training procedure as before[19].

First, the individual atomic embedding was concatenated and transformed into an initial atomic representation $\mathbf{h}_i^0$ with a multi-layer perceptron (MLP). The atom representations $\mathbf{h}_i^0$ were transformed through three layers of message passing. In each message-passing layer, the atomic representations were transformed via Eq. (1)

$$\mathbf{h}_i^{l+1} = \phi\left(\mathbf{h}_i^l, \sum_{j \in \mathcal{N}(i)} \psi\left(\mathbf{h}_i^l, \mathbf{h}_j^l, \mathbf{r}_{i,j,}\right)\right), \quad (1)$$

where $\mathbf{h}_i^l$ is the atomic representation of the *i*-th atom at the *l*-th layer; $j \in \mathcal{N}(i)$ is the set of neighboring nodes connected via edges; $\mathbf{r}_{i,j}$ the inter-atomic distance represented in terms of Fourier features, using a sine- and cosine-based encoding; $\psi$ is an MLP transforming node features into message features $\mathbf{m}_{ij}$: $\mathbf{m}_{ij} = \psi(\mathbf{h}_i^l, \mathbf{h}_j^l, \mathbf{r}_{i,j})$ for 3D graphs, and $\mathbf{m}_{ij} = \psi(\mathbf{h}_i^l, \mathbf{h}_j^l)$ for 2D graphs; $\sum$ denotes the permutation-invariant pooling Operator (*i.e.*, sum) transforming $\mathbf{m}_{ij}$ into $\mathbf{m}_i$: $\mathbf{m}_i = \sum_{j \in \mathcal{N}(i)} \mathbf{m}_{ij}$; and $\phi$ is an MLP transforming $\mathbf{h}_i^l$ and $\mathbf{m}_i$ into $\mathbf{h}_i^{l+1}$. The resulting atomic features of all layers [$\mathbf{h}_i^{l=1}$, $\mathbf{h}_i^{l=2}$, $\mathbf{h}_i^{l=3}$] were concatenated and transformed through an MLP, resulting in the final atomic features. The atom features were then pooled through a graph multiset transformer (GMT)[54] with four attention heads that produce an overall molecular feature vector.

This procedure was conducted for both input molecular graphs, where no weights were shared between the two GNN modules except for the initial embedding layers of the atom-level representations. The pooled molecular representations were then concatenated to a learned representation of the reaction conditions. This subsequent reaction representation was further transformed via a final MLP, converting the latent space into the desired reaction output. Both of the examined problems, namely, the prediction of reaction yield and the prediction of binary reaction outcome, were addressed as regression tasks. The output for reaction yield was defined within the range of floating values from 0 to 1, whereas for binary reaction outcomes, it was defined as 0 (non-reactive) or 1 (reactive). For each reaction, the total yield was calculated by summing the observed mono- and/or dialkylated species.

**Number of hyperparameters.** The number of features of the internal representation of the GTNN was 128, with the exception of the embedding dimension for the reaction and the atomic properties, which was set to 64. Additionally, the first MLP layer after multiset graph transformer-based pooling was configured to have 256 dimensions. The transformer architecture employed two attention heads for pooling. These parameter settings translated into neural network sizes with approximately two million trainable parameters for GTNN.

### Virtual library enumeration

Active molecules from the MAGL project containing a *N*-heteroaromatic handle for Minisci-type alkylation were filtered to 125 candidates with sufficient powder stock. In parallel, a unique list of all carboxylic acids available in-house at Roche and from Enamine (Monmouth Junction, NJ, USA) was assembled and filtered down to 211 acids (SI4.2 and Supplementary Data 1) based on a high fraction of sp³ carbon atoms and a molecular weight of lower than 230 g/mol. Using Minisci reaction templates for *N*-heteroaromatic rings, each of the 125 available project compounds were then virtually enumerated with the 211 carboxylic acids to generate 26,375 virtual compounds for processing through the multi-objective filtering funnel Fig. 1.

## Multi-template docking and 2D3D-scoring

The procedure for generating poses through multi-template pairing and 2D3D scoring has previously been described and the code has been made publicly available[40]. A total of 143 Roche internal co-crystal structures served as templates for docking. For each ligand, the template structure was selected on the basis of the maximum common substructure (MCS) between the ligand and the template ligand. The pose was generated with the GOLD docking program by constraining the ligand to the MCS part of the reference ligand[55]. The template docking approach was used as a data augmentation method to generate virtual protein-ligand complex structures for 3245 compounds with measured activities that serve as training data for the 2D3D scoring function, as described[40]. The 2D scoring model within the 2D3D scoring function is an attentive fingerprint model[31,56] trained on Roche in-house data. Using a four-fold cross-validation on random splits of a data set containing 7487 compounds with annotated biological activity to MAGL, this 2D model achieved a coefficient of determination ($R^2$) of 0.72 and a root mean square error (RMSE) of 0.77. The 3D scoring model relies on a frequency interaction ratio[40] and is taken without further adaptation from the original publication, representing a statistical model trained on data. The final "2D3D scoring" function is obtained by combining these 2D and 3D models into a single scoring metric. Subsequently, a virtual library of prospective ligands was subjected to multiple template docking, and the poses generated were ranked using the MAGL 2D3D scoring function.

## Synthesizability assessment

The enumerated library consisted of 26,350 potential molecules generated from 125 selected MAGL-inhibiting molecules featuring N-heteroaromatic fragments and 211 carboxylic acids. Each molecule was labeled as positive or negative on the basis of predicted reaction outcomes: molecules with a predicted reaction yield of ≥5% were classified as positive, while those with lower predicted yields were classified as negative. The final predictions for each molecule were based on the mean of three models, providing a robust assessment for subsequent machine learning model training and reaction property predictions.

## Absorption, distribution, metabolism, and excretion (ADME) property prediction

ADME property predictions were performed for 212 molecules identified as synthesizable and predicted to be active by multi-template docking. Attentive fingerprint models were used[31,56], which were trained on proprietary Roche internal data sets. The predicted ADME properties included LogD, lyophilisation solubility assay (LYSA) solubility in μg/mL, P-glycoprotein (P-gp) apparent permeability, and PAMPA in $10^{-6}$ cm/s. The four ADME endpoints (i.e., LogD, LYSA, P-gp, PAMPA) were predicted using multitask GNN models as detailed elsewhere[57].

## Synthesis of novel MAGL inhibitors

The 14 selected MAGL inhibitor molecules (**18**–**31**) were synthesized using a four-step procedure. The key step involved the alkylation of the benzoxazole building block (**9**) under Minisci-type conditions, which were pre-screened in silico and validated through HTE, yielding intermediates **9a**–**9i**. Subsequently, these intermediates were coupled with four different head groups (**13**–**16**) using classic amide coupling techniques. For this study, compound **17**, consisting of non-alkylated **9** and head group **13**, was freshly synthesized. All reactions were conducted under ambient conditions in glass reaction vessels equipped with pressure release caps and stirring bars. Purification was achieved via flash chromatography or reversed-phase high-performance liquid chromatography (RP-HPLC). Structural elucidation was performed using nuclear magnetic resonance (NMR) spectroscopy and high-resolution mass spectrometry (HRMS). The complete experimental procedures, analytical results, and spectra for all compounds are provided in the SI (SI5 and SI6).

## Biological characterization

**Measurement of IC$_{50}$ values.** The compounds were dissolved in DMSO at 10 mM and serially diluted with assay buffer to give final concentrations ranging from 12.5 μM down to 70 pM in the presence of MAGL protein. The dilutions were transferred to a 384-well assay plate containing a recombinant MAGL protein in assay buffer (50 mM TRIS, 1 mM EDTA, 0.01% Tween 20, v/v and 2.3% DMSO, v/v) and incubated at room temperature for 15 min. The reaction was initiated by adding the substrate 2-AG in assay buffer and incubation was conducted with gentle shaking at room temperature for 30 min. The final concentration of the MAGL protein was 50 pM, 8 μM for 2-AG, and 2.5% (v/v) DMSO. Upon completion of incubation, the reactions were quenched with a double assay volume of acetonitrile containing 4 μM d8-arachidonic acid (AA). The accumulation of d8-AA was monitored using an online solid phase extraction system (Agilent RapidFire) coupled to a triple quadrupole mass spectrometer (Sciex5000 or Agilent 6460). The samples were loaded in a C18 cartridge with 99% water/acetonitrile (v/v) and eluted with 5 mM ammonium acetate in 90% acetonitrile (v/v). A mass spectrometer was operated in the ESI mode with mass transitions m/z = 303.1-259.1 for AA and m/z = 311.1-267.1 for d8-AA, respectively. IC$_{50}$ values were fitted based on the ratio of AA / d8-AA intensities at different points of the serial dilution assay.

**X-ray crystal structures of MAGL complexes with 17, 23, 27 and 29.** The human MAGL protein (amino acid residues 1-303) with mutations Lys$^{36}$Ala, Leu$^{169}$Ser, and Leu$^{176}$Ser (Cepter Biopartners, Nutley, NJ, USA) was concentrated to 10.8 mg/mL. Crystallization tests were performed by sitting-drop vapor diffusion at 21 °C. Crystals appeared within two days out of 0.1 M MES pH 6.5, 6–13% PEG MME5K, 12% isopropanol. The crystals were soaked for 16 h in a crystallization solution with 10 mM ligand. For data collection, crystals were flash-cooled at 100 K with 20% ethylene glycol added as cryo-protectant to the soaking solution. The X-ray diffraction data were collected at a wavelength of 0.9999 Å using an Eiger2X 16M detector at the X10SA beamline of the Swiss Light Source (Villigen, Switzerland). Data were processed with XDS and scaled with SADABS (Bruker, Billerica, MA, USA). The crystals belong to the space group C2221 with cell axes a = 89.96 Å, b = 127.45 Å, c = 63.03 Å. They diffract to a resolution of 1.65 Å. The structure was determined by molecular replacement with PHASER using the coordinates of PDB entry 3PE6 as a search model[58]. The difference in electron density was used to place the compounds. The structure was refined with programs from the CCP4 suite and Buster. Model building was performed with COOT[59]. Data collection and refinement statistics are summarized in SI2.5.

## Reporting summary

Further information on research design is available in the Nature Portfolio Reporting Summary linked to this article.

# Data availability

**Training data:** The SURF-formatted experimental data set, containing 13,490 Minisci-type C-H alkylation reactions, has been made publicly available via Figshare (https://doi.org/10.6084/m9.figshare.28294850)[60]. **Co-crystal structures:** The coordinates of the MAGL co-crystal structure of the initial hit compound **17** have been deposited in the PDB under accession code 7PRM. The three co-crystal structures of optimized molecules (**23**, **27**, and **29**) are available under accession codes 9I5J, 9I9C, and 9I3Y, respectively. Source data are provided with this paper.

# Code availability

**Code:** A reference implementation of the geometric machine learning platform based on PyTorch[48] and PyTorch Geometric[47] is available at

https://github.com/ETHmodlab/minisci(rep. DOI: 10.5281/zenodo. 8344587, https://zenodo.org/record/8344587)[61].

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

## Acknowledgements

We thank the ADME team at F. Hoffmann-La Roche Ltd. for conducting various in vitro experiments and providing their valuable input as subject matter experts.

## Author contributions

Correspondence: R.E.M., U.G., G.S. D.F.N.: Conceptualization, data curation, formal analysis, investigation, methodology, supervision, creation of figures, plate design and condition selection, design and development of the ML-guided filtering funnel, experimental work, literature analysis, writing-original draft. K.A.: Conceptualization, data curation and analysis, formal analysis, investigation, methodology, initial software development, model training and validation, design and development of the ML-guided filtering funnel, creation of figures, writing-original draft. Y.S.: Experimental work, chemical synthesis, isolation and characterization of compounds 18-31, writing of the experimental procedure, writing-original draft. A.T.M.: Formal analysis, investigation, methodology, chemical space enumeration, analysis of the molecular properties and biological activity, chemical space selection, writing-original draft. A.Tos.: Methodology-validation, template-based docking and potency prediction of the in silico library of 26,375 virtual molecules. J.B.: Methodology-validation, co-crystallisation of molecules 17, 23, 27 and 29. H.B.: Supervision, project administration. M.B.: Methodology-validation, characterization of the final molecules by NMR and HRMS. A.H.: Methodology-validation, biological profiling, on-target, off-target ADMET characterization, of the molecules, writing-review and editing. D.H.: Methodology-validation, biological profiling. R.H.: Methodology-validation, biological profiling. C.K.: Supervision, project administration, writing-review and editing. M.R.: Methodology-validation, development of the 2D-based potency prediction model. P.S.: Methodology-validation, on- and off-target target prediction. T.S.: Methodology-validation, conduction of the activity-based protein profiling experiments. A.Top.: Methodology-validation, biological profiling. A.W.: Methodology-validation, co-crystallisation of molecules 17, 23, 27 and 29. M.B.W.: Methodology-validation, ADMET characterization of the molecules. J.W.: Methodology-validation, experimental synthesis work. B.K.: Formal analysis, investigation, methodology, writing-review and editing, selection of the 125 hit molecules and the final compounds. M.v.S.: Formal analysis, investigation, methodology, supervision for the activity-based protein profiling experiments. R.E.M.: Formal analysis, investigation, methodology, supervision, project administration, writing-original draft. U.G.: Formal analysis, investigation, methodology, supervision, project administration, writing-original draft. G.S.: Formal analysis, investigation, methodology, supervision, project administration, writing-original draft.

## Funding

## Competing interests

G.S. and P.S. declare a potential financial conflict of interest as co-founders of inSili.com LLC, Zurich, and Xanadys LLC, Zurich. D.F.N., K.A., Y.S., A.T.M., A.T., J.B., H.B., M.B., A.H., D.H. R.H., C.K., M.R., T.S., A.T., A.W., M.B.W., J.W., B.K., R.E.M. and U.G. are full employees of F. Hoffmann-La Roche Ltd.
