## [Transparent Peer Review file · Nature Communications]

Expediting hit-to-lead progression in drug discovery through reaction prediction and multi-dimensional optimization

Corresponding Author: Professor Gisbert Schneider

Version 0:

Reviewer comments:

Reviewer #1

(Remarks to the Author)

Nippa et al. showcase an ML-backed platform for utilizing their specialization in Minisci reactions to find compounds with sub micromolar potency towards monoacylglycerol lipase (MAGL). This publication builds upon their work in high throughput experimentation and machine learning, specifically GNNs, and is a nice representation of how to combine ML and experimental techniques to arrive at promising bioactive leads. I recommend publication of this manuscript with minor revisions.

Specifically:

1. More detail on how which (hereo)aryl + acid compounds were selected for the experimental 13K HTE screen. Was there particular care given to specific locations of chemical space?
2. The multi-objective filtering is a critical step. What method(s) were used to determine the predicted values for the physicochemical properties (LogD, LYSA, P-gp, PAMPA) and how where they determined accurate enough for this filtering step?
3. The GNN was assessed via reaction yield and binary outcome prediction. Does binary outcome prediction refer to classifying yields as above/below a certain threshold? How were mixtures of mono- and/or di-alkylated species dealt with in yield predictions?
4. Figure 4A is difficult to read, mostly because the starting pyridine is in the bottom left of the figure. Moving the structures so that the overall transformation (compound 1 -> compound 9) is more obvious would help.

Reviewer #2

(Remarks to the Author)

This manuscript describes 1) development of a machine learning algorithm for Minisci alkylations using diverse carboxylic acids and heteroarenes (~13000 reactions) and 2) the use of these machine learning models for a medicinal chemistry campaign. In the medicinal chemistry campaign the machine learning model for minisci alkylations was used to predict reaction success for the targets. The targets were further triaged based on standard medicinal chemistry strategies like structure based design or physicochemical properties and the predicted potency.

Overall this is a nice medicinal chemistry story. As mentioned in the manuscript, machine learning models for minisci alkylations has previously been reported (reference 19). Although the structural diversity of the dataset used to build the Minisci reaction predictive models is significantly higher in this new manuscript, the manuscript lacks sufficient novelty to merit publication in Nature communications.

Reviewer #3

(Remarks to the Author)

The manuscript by D. Nippa et al. describes optimization of monoacylglycerol lipase (MAGL) inhibitors using "multi-objective

molecular optimization". The authors employed Minisci-type C-H alkylation reactions between N-heteroaromatic compounds and carboxylic acids for derivatization and optimization of lead compounds. To predict synthetic accessibility of potential products, the authors have generated experimental data evaluating synthesis yields for 13,490 reactions varying not only synthetic blocks (80 aromatic fragments and 59 carboxylic acids) but also reaction conditions, nature of the oxidant and acid additive. The data was used for GNN training in predicting reaction yields for lead derivatives. For the design of improved inhibitors, the authors have generated a virtual library from 125 hits with proven anti-MAGL activity and 211 carboxylic acids. The resulting library of 26,375 compounds was subjected to multi-template docking using 143 Roche internal co-crystal structures yielding 1675 molecules with predicted potency in nanomolar range. The data from GNN training of experimental yields for Minisci model reactions allowed to predict 668 compounds that could be synthesized with reasonable yields. Out of this pool, 212 molecules were predicted to have IC50 in low nanomolar range and were further filtered based on physicochemical properties and ADME parameters. 34 candidates have been synthesized on a miniature scale and 14 best-yielding reactions have been chosen for scaling up, although all 34 produced yields of more than 5% as predicted. Six out of 14 compounds have shown desired potency in 0.1-10 nM range. Pearson correlation between experimental and predicted IC50 values was =0.52. Binding modes for three top compounds were confirmed by solving the x-ray structures of the co-crystals.

The manuscript aims at addressing an important problem in drug discovery: minimization of the chemical synthesis use to reduce the cost and speed up the process. Significant efforts are being made to utilize different approaches for the prediction of synthetic accessibility of virtual compounds. The biggest problem in utilizing AI-based approaches is lack of usable data for training sets. The valuable feature of the manuscript is generation of a dataset for a useful reaction that can be used in optimization of many drug leads and making it public. The authors provide an example that suggests that the methodology can work, although it is unclear how much advantage over other approaches like utilization of large virtual libraries it provides. One can guess that it saves computational resources, but it should be discussed in the paper. The timeline would also be helpful as speeding up drug discovery is the aim of many in the field.

The limitations and drawbacks of the method should also be addressed. For example, correlation coefficient between predicted and experimental IC50 for compounds that have been synthesized is only 0.52, which is lower than desired and achievable by other methods. What could be done to improve it?

In general, the manuscript is well written. However, the abstract could be worded better, so that general public can appreciate the goal and the value of the studies. The objectives like "significant", "effectively", "valuable" are overused without providing clarity.

Easily correctable error/missed step is in "Material and Methods" in "Measurement of IC50 values" that state that the compounds serially diluted in DMSO to result in concentrations covering the compound activity range (usually between 12.5 uM and 70 pM) were transformed to assay plates containing MAGL protein. Were these concentrations of the stocks or final in the reaction mixtures?.

The rest of the methods appear to be described in sufficient details and are appropriate for the studies.

Version 1:

Reviewer comments:

Reviewer #1

(Remarks to the Author)

The authors have addressed all my concerns. I recommend publication.

Reviewer #3

(Remarks to the Author)

Noted deficiencies of the manuscript have been corrected. It can now be published "as is".

Revision NCOMMS-25-07758

Reviewer #1 (Remarks to the Author):

Nippa et al. showcase an ML-backed platform for utilizing their specialization in Minisci reactions to find compounds with sub micromolar potency towards monoacylglycerol lipase (MAGL). This publication builds upon their work in high throughput experimentation and machine learning, specifically GNNs, and is a nice representation of how to combine ML and experimental techniques to arrive at promising bioactive leads. I recommend publication of this manuscript with minor revisions.

1. More detail on how which (hereo)aryl + acid compounds were selected for the experimental 13K HTE screen. Was there particular care given to specific locations of chemical space?

>>> We now detail our building-block (BB) selection workflow, designed to (i) capture medicinal-chemistry-relevant chemical space, (ii) ensure compatibility with Minisci-type radical alkylation, and (iii) guarantee practical availability for high-throughput experimentation.

The 80 N-heteroarenes (F1–F80) and 59 carboxylic acids (A1–A59) were selected via a three-step workflow:

- 1) Compatibility filter: Candidates with functional groups known to interfere with Minisci alkylation (e.g., aldehydes, free thiols) were removed.
- 2) Commercial availability: Only compounds available in gram quantities from at least one vendor were retained.
- 3) Manual diversity pick: From the remaining compounds, we manually selected heteroarenes covering common medicinal-chemistry scaffolds (e.g., pyridines, pyrimidines, indoles), sp³-rich acids spanning linear, branched, cyclic, heterocyclic, and aromatic motifs.

The final lists are presented in SI 4.1 (F1–F80) and SI 4.2 (A1–A59).

The following paragraphs were inserted to the Manuscript

“Using the 24-well plate design (Figure 3F), we evaluated Minisci-type alkylations between a curated set of 80 commercially available N-heteroarenes (comprising pyridines, pyrimidines, and indoles, labeled F1–F80 in SI 4.1) and 59 sp³-rich carboxylic acids (encompassing linear, branched, cyclic, heterocyclic, and aromatic motifs, labeled A1–A59 in SI 4.2). Candidates were first filtered to remove incompatible functional groups. They were then required to be in stock (greater than 1 g) from at least one supplier, and finally, manually down-selected to maximize scaffold diversity. The reactions were set up using automated liquid handling of prepared stock solutions, and then executed in glass vials on a parallel screening plate under a normal atmosphere, with heating and stirring.”

2. The multi-objective filtering is a critical step. What method(s) were used to determine the predicted values for the physiochemical properties (LogD, LYSA, P-gp, PAMPA) and how were they determined accurate enough for this filtering step?

>>> Reaction prediction and potency prediction were applied as hard cut-offs, reducing the initial library from approximately 26,000 to 212 candidates. The four ADMET properties (LogD, LYSA, P-gp, PAMPA) were then used only as guidance to identify and consider particularly extreme profiles when selecting the final 14 molecules from those 212. No fixed thresholds were imposed on any ADMET endpoint to avoid error propagation across models. Reaction feasibility and potency were most critical for our objectives, ultimately yielding a set of 14 synthesizable compounds with improved activity.

All ADMET predictions were generated using the multitask GNN models described in Napoli et al. (2025) ("multi task GNN models for ADMET prediction"; see Methods).

The following paragraphs were inserted to the Manuscript

Results:

"The reaction and potency predictions served as strict criteria, narrowing the virtual library to 212 molecules. The final selection of 14 compounds was guided by calculated ADME parameters, using an approach that avoided fixed thresholds to minimize the propagation of model error through sequential filtering steps."

Methods:

"four ADME endpoints (i.e., LogD, LYSA, P-gp, PAMPA) were predicted using multitask GNN models as detailed elsewhere [57]"

3. The GNN was assessed via reaction yield and binary outcome prediction. Does binary outcome prediction refer to classifying yields as above/below a certain threshold?

>>> Binary outcome prediction classifies each reaction as positive or negative based on whether its predicted yield meets a predefined threshold ($\geq 5\%$).

The following passages were added to the manuscript:

Introduction

"Data analysis revealed that 30% of the reactions resulted in positive reaction outcomes, i.e., successful alkylation of the heterocyclic fragment with a detectable reaction yield of $\geq 5\%$ (Figure 3D)."

Figure 3D

“Predicted reaction outcomes: Molecules with a predicted reaction yield $\geq 5\%$ are classified as positive, otherwise negative.”

Results

“The application of the GNN reaction prediction model to validate the synthetic feasibility resulted in 668 compounds with a predicted reaction yield of $\geq 5\%$ (Figure 4D), confirming a positive, binary reaction outcome.”

Methods

“The enumerated library consisted of 26,350 potential molecules generated from 125 selected MAGL-inhibiting molecules featuring N-heteroaromatic fragments and 211 carboxylic acids. Each molecule was labeled as positive or negative on the basis of predicted reaction outcomes: molecules with a predicted reaction yield of $\geq 5\%$ were classified as positive, while those with lower predicted yields were classified as negative.”

How were mixtures of mono- and/or di-alkylated species dealt with in yield predictions?

>>> We predicted the binary reaction outcome (Does the reaction work: Yes/No?), so the models were trained using the sum of mono- and/or di-alkylated species. To prevent misunderstandings, we added two statements to the main manuscript: one in the results section and one in the discussion section:

Results:

“The application of the GNN reaction prediction model to validate the synthetic feasibility resulted in 668 compounds with a predicted reaction yield of $\geq 5\%$ (Figure 4D), confirming a positive, binary reaction outcome.”

Discussion:

“The reaction prediction models were then used to assess the synthetic feasibility of these candidates (binary “yes/no” reaction outcome), identifying 668 molecules with predicted reaction yields of $\geq 5\%$.”

Furthermore, the methods section received an additional sentence: “For each reaction, the total yield was calculated by summing the observed mono- and/or di-alkylated species.”

4. Figure 4A is difficult to read, mostly because the starting pyridine is in the bottom left of the figure. Moving the structures so that the overall transformation (compound 1 \rightarrow compound 9) is more obvious would help.

>>> Thanks for the suggestions. We've adjusted Figure 4A to make compound 1 more prominent in the upper left corner.

Reviewer #2 (Remarks to the Author):

This manuscript describes 1) development of a machine learning algorithm for Minisci alkylations using diverse carboxylic acids and heteroarenes (~13000 reactions) and 2) the use of these machine learning models for a medicinal chemistry campaign. In the medicinal chemistry campaign the machine learning model for minisci alkylations was used to predict reaction success for the targets. The targets were further triaged based on standard medicinal chemistry strategies like structure based design or physicochemical properties and the predicted potency.

Overall this is a nice medicinal chemistry story. As mentioned in the manuscript, machine learning models for minisci alkylations has previously been reported (reference 19). Although the structural diversity of the dataset used to build the Minisci reaction predictive models is significantly higher in this new manuscript, the manuscript lacks sufficient novelty to merit publication in Nature communications.

>>> We appreciate the reviewer's feedback and acknowledge prior machine learning models for Minisci alkylations, including our own work (reference 19). However, we respectfully assert that the present manuscript offers substantial novelty, primarily due to its unique and powerful integration of diverse tools and strategies across the drug discovery pipeline. The key differentiating aspects and significant novelty of this study, beyond just Minisci reaction prediction, include:

- **Holistic, integrated workflow for expedited hit-to-lead progression:** Unlike previous work focusing primarily on reaction prediction, this study presents a fully integrated medicinal chemistry workflow that effectively diversifies hit and lead structures to accelerate the hit-to-lead optimization phase. This is not merely an incremental improvement in a single tool, but a synergistic combination of methodologies that enables a rapid optimization cycle, as demonstrated by the timeline of approximately one month for computational work, synthesis, and biological measurement. This comprehensive approach is a significant step towards streamlining early drug discovery by reducing DMTA cycle times.
- **Multi-Dimensional Optimization (MDO) funnel:** A cornerstone of the novelty is the sophisticated "machine learning funnel" employing a multi-dimensional optimization strategy. This funnel simultaneously considers:
 - **Structure-based scoring for potency:** Utilizing multi-template docking and a 2D3D-hybrid scoring function to predict and prioritize compounds with improved on-target potency. This goes beyond general reactivity prediction to directly address a critical medicinal chemistry optimization parameter.

- **Validated synthetic feasibility prediction:** While reaction prediction was reported before, its *integration* into a prospective design pipeline for hit-to-lead optimization, ensuring that only synthetically accessible compounds are pursued, is crucial and novel. The study explicitly demonstrates how this filtering step can reduce the number of compounds to be synthesized, thereby saving time and resources.
 - **Comprehensive ADME property prediction:** Incorporating predictions for key physicochemical and ADME properties (expressed in terms of LogD, solubility, P-gp permeability, PAMPA) enhances the likelihood of obtaining a favorable pharmacokinetic profile from the outset. This multi-objective optimization, balancing potency, synthesizability, and ADME, is critical for drug development and represents an advancement over isolated predictive models.
- **Application to late-stage functionalization (LSF) with demonstrated efficiency:** Our study specifically focuses on Minisci-type C-H alkylations for LSF, a relevant strategy for diversifying complex molecules without extensive multi-step synthesis. The practical demonstration of a shortened synthesis route (from seven steps to three for the alkylated intermediate) directly translates to enhanced synthesis efficiency and reduced cycle times, an important outcome for medicinal chemistry. This practical application of ML-guided LSF to deliver improved compounds is a key novel contribution.
 - **Rigorous experimental validation and impact:** Our study culminates in the successful synthesis and biological characterization of 14 novel MAGL inhibitors, demonstrating substantial potency improvements (up to 4500-fold) and favorable pharmacological profiles. Furthermore, co-crystallization studies provide atomic-level insights into the binding modes of the new inhibitors, validating the structure-based design component of the workflow. This end-to-end demonstration, from computational design to validated *in vitro* and structural data, significantly elevates the impact beyond a mere methodological report.
 - **Emphasis on FAIR data principles and reproducibility:** The explicit mention and implementation of FAIR (Findable, Accessible, Interoperable, Reusable) principles for data curation and sharing (e.g., SURF format data, public code, co-crystal structures upon acceptance) is a crucial aspect for advancing machine learning in chemistry. The SURF data set contains 13,500 novel Minisci reactions directly usable for machine learning applications. This commitment to open science practices distinguishes the work and fosters future research.

While our prior work laid the methodological groundwork for Minisci reaction prediction, the current manuscript moves far beyond that by integrating this prediction into a comprehensive, prospectively validated drug discovery pipeline. The combination of all these distinct tools – C-H functionalization, multi-objective optimization across hit-to-lead and lead optimization, synthesis

success prediction, and a strong structure-based component – culminates in a highly effective and innovative approach to drug discovery. This holistic, data-driven, and experimentally validated workflow represents a substantial advance in automated molecular design.

Reviewer #3 (Remarks to the Author):

The manuscript by D. Nippa et al. describes optimization of monoacylglycerol lipase (MAGL) inhibitors using “multi-objective molecular optimization”. The authors employed Minisci-type C-H alkylation reactions between N-heteroaromatic compounds and carboxylic acids for derivatization and optimization of lead compounds. To predict synthetic accessibility of potential products, the authors have generated experimental data evaluating synthesis yields for 13,490 reactions varying not only synthetic blocks (80 aromatic fragments and 59 carboxylic acids) but also reaction conditions, nature of the oxidant and acid additive. The data was used for GNN training in predicting reaction yields for lead derivatives. For the design of improved inhibitors, the authors have generated a virtual library from 125 hits with proven anti-MAGL activity and 211 carboxylic acids. The resulting library of 26,375 compounds was subjected to multi-template docking using 143 Roche internal co-crystal structures yielding 1675 molecules with predicted potency in nanomolar range. The data from GNN training of experimental yields for Minisci model reactions allowed to predict 668 compounds that could be synthesized with reasonable yields. Out of this pool, 212 molecules were predicted to have IC₅₀ in low nanomolar range and were further filtered based on physicochemical properties and ADME parameters. 34 candidates have been synthesized on a miniature scale and 14 best-yielding reactions have been chosen for scaling up, although all 34 produced yields of more than 5% as predicted. Six out of 14 compounds have shown desired potency in 0.1-10 nM range. Pearson correlation between experimental and predicted IC₅₀ values was =0.52. Binding modes for three top compounds were confirmed by solving the x-ray structures of the co-crystals.

The manuscript aims at addressing an important problem in drug discovery: minimization of the chemical synthesis use to reduce the cost and speed up the process. Significant efforts are being made to utilize different approaches for the prediction of synthetic accessibility of virtual compounds. The biggest problem in utilizing AI-based approaches is lack of usable data for training sets. The valuable feature of the manuscript is generation of a dataset for a useful reaction that can be used in optimization of many drug leads and making it public.

The authors provide an example that suggests that the methodology can work, although it is unclear how much advantage over other approaches like utilization of large virtual libraries it provides. One can guess that it saves computational resources, but it should be discussed in the paper. The timeline would also be helpful as speeding up drug discovery is the aim of many in the field.

>> We appreciate the reviewer's insightful comments regarding the advantages of our integrated workflow and the importance of demonstrating its efficiency. We agree with the suggestions made. Our methodology, which combines miniaturized high-throughput

experimentation (HTE) with deep learning and multi-objective molecular optimization, offers several distinct advantages over simply utilizing large virtual libraries. While large virtual libraries allow for vast exploration of chemical space, their sheer size often presents a significant bottleneck in downstream experimental validation. The key differentiating factors of our approach are:

1. **Synthetic feasibility as an early filter:** A major limitation of purely virtual library enumeration is the synthesis bottleneck. Many computationally designed molecules, even if predicted to be potent, are challenging or impossible to synthesize efficiently. Our workflow integrates reaction prediction models, trained on a comprehensive HTE-generated dataset of Minisci-type C-H alkylation reactions, as an early and crucial filter. This reduces the number of "undesirable" candidates that would otherwise progress to the synthesis stage, saving considerable time, material, and cost. For instance, out of 26,375 enumerated virtual molecules, only 668 were predicted to be synthesizable with a yield of $\geq 5\%$. This direct incorporation of synthetic accessibility as a design criterion differentiates our approach from methods that solely rely on property prediction and thus offer an advantage over other approaches, like utilization of large virtual libraries. This strategy also points to the future direction we believe the field should take in the coming years.
2. **Fast optimization cycle:** By combining miniaturized HTE, which allows for rapid identification and optimization of reaction conditions with minimal material consumption, with our computational funnel, we achieve a fast optimization cycle. This is exemplified by the complete hit-to-lead optimization process being completed within one month, encompassing computational work (1 day), efficient synthesis (2 weeks), and biological measurement (2 weeks). This rapid turnaround greatly accelerates the drug discovery process compared to iterative experimental cycles often required when relying solely on large virtual libraries and traditional synthesis routes. While large virtual libraries offer breadth, our methodology provides a more targeted and efficient path by integrating synthetic accessibility and multi-objective optimization early in the design process, leading to a faster and more resource-effective hit-to-lead progression.

We have added the following paragraphs in the discussion to reflect the points above in the manuscript.

The Paragraph:

“The reaction forward prediction models demonstrated their robustness not only by accurately predicting reaction outcomes for known substrates, as seen with the random split (*i.e.*, 0D), which achieved a mean absolute error (MAE) of 6.7%, high correlation ($r=0.83$) and a precision value of 90.9%, but also by successfully extrapolating to novel chemical space. When applied to unseen carboxylic acids and N-arenes independently (1D split) and combined (2D split), the models maintained robust performance, with MAEs in the range of 11.4-12.6%, precision values up to 85.4%, and correlation coefficients ranging from $r=0.51$ to $r=0.55$. Recent studies have shown that combining 2D and 3D scoring methods achieves state-of-the-art performance, further corroborating the effectiveness of our approach. These results demonstrate the ability of the model to

exhibit some degree of generalization beyond the molecules in the training data, providing valuable predictions even in unexplored regions of the reaction space].

was changed to:

“The reaction prediction models demonstrated their robustness not only by accurately predicting the reaction outcome for known substrates, as seen with the random data split (OD split), which achieved a mean absolute error (MAE) of approximately 7%, high correlation ($r = 0.83$), and a precision value of 91%. When applied to unseen carboxylic acids and N-arenes, either independently (1D split) or combined (2D split), the models maintained robust performance, with MAEs in the range of 11–13%, precision values up to 85%, and correlation coefficients ranging from $r = 0.51$ to $r = 0.55$. This ability to partially extrapolate to novel chemical space, combined with the integration of predicted synthetic feasibility, provides an advantage over relying solely on the enumeration and screening of large virtual libraries, where synthetic accessibility is often a downstream challenge [27].”

Further, we provided a more detailed discussion of the timelines by adding:

“The full optimization cycle was completed in one month. The computational work, including virtual library enumeration, multi-template docking, reaction prediction, and ADMET property prediction, was completed in approximately one day. Compound synthesis, using miniaturized HTE and scale-up, took two weeks. The biological measurements were also completed in two weeks. This condensed timeline demonstrates how integrating computational and experimental workflows from the outset outpaces traditional drug discovery pipelines that do not include predictive modeling.”

Additionally, two sentences were added to the summary:

“It provides a distinct advantage over simply enumerating and screening vast virtual libraries, primarily because exhaustive library enumeration often presents considerable synthetic challenges in laboratory experiments. Importantly, the applicability and success of our workflow hinge on the availability of high-quality, well-curated data sets for machine learning and access to reliable binding models for computational template-based docking and affinity prediction.”

The limitations and drawbacks of the method should also be addressed. For example, correlation coefficient between predicted and experimental IC₅₀ for compounds that have been synthesized is only 0.52, which is lower than desired and achievable by other methods. What could be done to improve it?

>> Thank you very much for this important point. Although our potency predictions correlate with experiment at $r = 0.52$, we would like to highlight that six inhibitors demonstrated a 45- to

4500-fold increase in potency compared with the template, achieving the desired potency improvements.

Physics-based methods (e.g., FEP and semi-empirical quantum chemistry) offer high-fidelity ΔG estimates but are too computationally costly for large-scale screening of tens of thousands of ligands. On the other hand, deep-learning approaches (e.g., GNNs) enable rapid scoring across vast libraries but still lack optimal scoring functions. To balance speed and reliability, we employed template-based docking with RF-scoring (Tosstorff et al. 2022), which has demonstrated both fast and accurate potency prediction.

To discuss the challenges of potency prediction and situate our results within the broader literature, we have added the following paragraph to the Discussion:

“The moderate correlation ($r = 0.52$) of the potency predictions with the experimental results reflects an inherent trade-off between computational cost and accuracy. Free energy perturbation [35] or semi-empirical quantum chemistry approximations [36, 37] could potentially refine ΔG estimates for a small subset of ligands but are impractical for large-scale screening. In contrast, deep-learning models enable rapid scoring of large molecule libraries, yet remain limited by the quality of their scoring functions. Here, we adopted template-based docking with RF-scoring [40], which has been shown to efficiently enrich active compounds. Developing next-generation scoring functions that combine the fidelity of physics-based methods with the throughput of machine learning models represents an opportunity for in silico hit and lead profiling.”

In general, the manuscript is well written. However, the abstract could be worded better, so that general public can appreciate the goal and the value of the studies. The objectives like “significant”, “effectively”, “valuable” are overused without providing clarity.

Easily correctable error/missed step is in “Material and Methods” in “Measurement of IC₅₀ values” that state that the compounds serially diluted in DMSO to result in concentrations covering the compound activity range (usually between 12.5 μ M and 70 pM) were transformed to assay plates containing MAGL protein. Were these concentrations of the stocks or final in the reaction mixtures?.

The rest of the methods appear to be described in sufficient details and are appropriate for the studies.

>> Thank you very much for these constructive suggestions.

Abstract clarity:

We removed vague qualifiers and focused on concrete goals and outcomes that a general reader can appreciate. For example, instead of saying “significant” or “valuable,” we now specify the scale and impact of our work.

Measurement of IC₅₀ values concentrations:

These concentration ranges refer to final assay concentrations in the reaction wells. We have updated the Materials and Methods accordingly:

“The compounds were dissolved in DMSO at 10 mM and serially diluted with assay buffer to give final concentrations ranging from 12.5- μ M down to 70-pM in the presence of MAGL protein.”

This makes explicit that the 12.5 μ M–70 pM range applies to the assay mixture rather than stock solutions.